# Synthesis and Characterization of Electrospun Carbon Nanofibers from Polyacrylonitrile and Graphite Nanoplatelets

**DOI:** 10.3390/ma16041749

**Published:** 2023-02-20

**Authors:** Hani Manssor Albetran

**Affiliations:** 1Department of Physics, College of Science, Imam Abdulrahman Bin Faisal University, P.O. Box 1982, Dammam 31441, Saudi Arabia; halbatran@iau.edu.sa; Tel.: +966-13-827-4155; Fax: +966-13-826-9936; 2Basic and Applied Scientific Research Center (BASRC), Imam Abdulrahman Bin Faisal University, P.O. Box 1982, Dammam 31441, Saudi Arabia

**Keywords:** carbon, electrical conductivity, electrospinning, graphite structure, nanofiber

## Abstract

Sol-gel electrospinning process was used to prepare electrospun carbon nanofibers (ECNFs) from polyacrylonitrile and graphite nanoplatelets. The nanofibers of as-electrospun carbon were calcinated in argon from room temperature to 500 °C for 1h. Scanning and transmission electron microscopy with energy-dispersive X-ray spectroscopy and X-ray diffractometry (XRD) were used to characterize the synthesized ECNFs. The smooth ECNFs with a diameter of 129 ± 43 nm comprised conical platelets of 30–200 µm length. The plane-layered nanofibers contained crystallites along the long fiber axis and were mainly parallel.

## 1. Introduction

Among the various types of nanomaterials, the excellent chemical, thermal, and electrical resistance and high strength-to-weight ratio make carbon nanofibers (CNFs) attractive for use in scanning probe microscope tips, sensors structural materials, hydrogen storage, field-emission displays, and nanometer semiconductor devices [1,2,3,4,5,6].

Electrospinning with heating in an inert atmosphere, vapor-grown carbon fibers, and plasma-enhanced chemical vapor deposition are three common approaches for CNF production. The CNFs produced by two traditional methods (vapor growth and plasma-enhanced chemical vapor deposition) involve a complicated process and has an associated high cost [7]. However, carbon nanofiber (CNF) production by electrospinning is simple and cost-effective for industrial use [8], and continuous submicron production (two to three orders of magnitude smaller than conventional CNFs) has become the predominant synthesis technique by using polymeric precursor fiber spinning and subsequent thermal treatment [9]. Furthermore, the electrospinning method is a process involving applied physics, polymer science, fluid mechanics, chemical, mechanical, electrical, and material engineering [7]. A syringe pump, a high voltage source, a syringe with a conductive needle, a conductive collector, a sol-gel of polymer (binder), copper wires, and a precursor are the components of the electrospun experiment to create nanofibers [10,11]. The fiber morphology is controlled by the experimental parameters and is dependent upon three processing parameter sets (sol-gel solution, electrospinning settings, and atmospheric environment) [12].

Firstly, the sol-gel solution is part of the initial set of altered parameters. Electrical conductivity, polymer concentration, surface tension, viscosity, and molecular weight are some of its characteristics. They are connected, and these connections have a significant impact on the diameter of electrospun nanofibers [13]. The molecular weight and concentration of the material solution, such as the polymer, solvent, and material source, affect the viscosity of the mixture. One of the most important variabilities in regulating bead and fiber diameter is the polymer content [13,14,15]. A higher concentration in a precursor solution results in a larger fiber diameter [16,17].

Secondly, electrospinning conditions, which include needle tip-to-collector distance, needle size, applied voltage, and flow rate [13]. The fiber diameter is significantly influenced by the optimally applied voltage [14,17,18,19]. Low nanofiber diameter is caused by high applied voltage (strong electrical repulsive forces), which causes fibers to be greatly stretched and lengthened [18]. Because it controls the amount of sol-gel solution available to be stretched into nanofibers, flow rate plays one of the most crucial roles in regulating the fiber diameter and bead formation [13,17,18]. The Taylor cone and nanofiber oscillation during electrospinning are influenced by the size and shape of the needle tip [20]. According to Ksapabutr et al. [20], Taylor cones with a sawtooth needle shape can be longer than their flat and regular equivalents. A needle tip-to-collector distance with a strong enough electric field gradient results in fibers with fewer bead defects, while a longer distance results in a larger nanofiber diameter as the electric field gradient gradually reduces [14]. High voltage setting, a long needle tip-to-collector distance, a low flow rate, and a relatively low concentration of polymer solutions limit the variation in product quality of electrospun fiber mats [14].

Finally, the atmospheric environment such as temperature, pressure, and humidity. With rising air temperature and falling atmospheric humidity, the average nanofiber diameter decreases [21].

Inagaki et al. [8] summarized CNF preparation by electrospinning. High-temperature carbonization of electrospun precursor polymers, including polyacrylonitrile (PAN) and copolymers (co-PAN), polyamic acid poly(vinyl alcohol), polybenzimidazole, and BBB, can be used to prepare CNFs [1,22,23,24,25,26,27]. The high melting point and carbon yield of PAN make it a common polymer for electrospun CNF production, yielding stable thermal products. The PAN-based ECNF surface can be modified by coating or activation to prepare up to a centimeter-long, continuous, and well aligned CNFs for polymer stiffening, toughening, and strengthening [4,28,29]. Multiwalled carbon nanotubes and coal-embedded PAN nanofibers are prepared by electrospinning [28,30]. CNF-reinforced composites application is limited by precursor and subsequent ECNF weak mechanical properties, complex synthesis, process difficulties, and cost [22,31]. Therefore, the selection of carbon precursors with suitable structures and nanotexture control, electrospinning parameter optimization and stabilization, carbonization, activation, and high-temperature heat treatment require additional study. The pore structure, extent of graphitization, electrical conductivity, and CNF metallic species loading are critical electrospinning and carbonization parameters and are dependent on the precursor selection (carbon, metal, and additives) and CNF preparation conditions.

In this work, polyacrylonitrile (PAN) and graphite nanoplatelets (GNPs) were used to prepare ECNFs by electrospinning and carbonization in argon at low temperature (500 °C). This synthesis was achieved for the first time to the best of the author’s knowledge. ECNF characterization included scanning and high-resolution transmission electron microscopy (SEM and HR-TEM, respectively), energy dispersive X-ray (EDS) spectroscopy, and X-ray diffractometry (XRD).

## 2. Experimental Methodology

### 2.1. CNF Reagents

PAN (*M_w_* 150,000 g/mol, 100% pure, Sigma-Aldrich, St. Louis, MO, USA), acetic acid (*M_w_* 60.05 g/mol, 99.7% pure, Sigma-Aldrich), N,N-dimethylformamide, ethanol (*M_w_* 46.07 g/mol, 99.5% pure, Sigma-Aldrich), and GNPs were used. The GNP synthesis has been described previously [32]; in brief, GNPs were prepared by ultrasonic exfoliation of graphite in a bath (40 kHz, 5 h, room temperature, in 70% isopropyl alcohol and 30% water).

### 2.2. Synthesis of Carbon Nanofibers

A carbon suspension of GNPs was prepared by mixing pulverized and sieved (200-mesh) GNP powder (2 g) with acetic acid (33.3 wt%) and ethanol (66.6 wt%). The solution was sonicated at room temperature for 30 min. PAN (0.6 g) was dissolved in N,N-dimethylformamide (10 mL) with continuous magnetic stirring at 50 °C for 60 min. The GNP and PAN solutions were mixed magnetically overnight at room temperature to prepare a homogeneous carbon sol–gel precursor solution for the electrospinning experiments (Figure 1).

Prior to being loaded into a 10 mL plastic syringe with a 25 gauge stainless steel needle (~0.514 mm inner diameter, 14 cm needle tip-to-collector distance), the ECNF sol–gel solution precursor was stirred ultrasonically for 5 min. Electrospun fiber mats were prepared by using a commercial Nabond^®^ electrospinning unit (Nabond Technologies Co., Ltd., Shenzhen, China) with a 2 mL/h solution flow rate that was controlled by a syringe pump. The effect of humidity is not a significant factor, as the electrospinning experiments were performed in a sealed environmental chamber at a constant temperature of 30 °C, which was maintained using a lamp heater. A 23 kV voltage was applied between the aluminum foil-covered collector and needle. The as-spun ECNFs were removed from the aluminum foil.

### 2.3. ECNF Stabilization and Carbonization

ECNFs were heated non-isothermally in air from room temperature to 250 °C in a furnace for 5 h at 5 °C/min, with a 1-h hold time at the peak temperature. Sample carbonization was undertaken in argon for 1 h in a high-temperature tube furnace at a peak 500 °C in a single step with a 5 °C/min heating rate.

### 2.4. Material Characterization

#### 2.4.1. SEM

SEM samples were prepared by direct fiber deposition on carbon double-tape-covered aluminum stubs with excess material removal. A ThermoFisher Teneo SEM (Waltham, MA, USA) (5 keV acceleration voltage, 0.1 nA beam current, at the Core Labs of King Abdullah University of Science and Technology) with a field emission gun, EDAX SDD detector for X-ray spectroscopy analysis, and secondary scattered electron imaging from in-lens T2 and T1 detectors were used. SEM-EDX chemical analysis was conducted at a 15 keV acceleration voltage and 1.6 nA beam current, with a good signal/noise ratio and heavy element excitation by X-ray emission. Oxford Instruments INCA (version 5.05) software was used for qualitative and quantitative EDS analyzes that were collected at a small point. The ECNF quantitative EDS analysis was approximated because of the large EDS interaction volume, which has a greater influence than fiber diameter.

#### 2.4.2. HR-TEM

ECNFs were dispersed with lacey-carbon-coated copper grids with a 200-mesh spacing. TEM and HR-TEM imaging was achieved using a ThermoFisher Titan ST microscope (Core Labs of King Abdullah University of Science and Technology) with a field emission gun (300 keV) in bright-field mode, a spatial resolution of ~0.18 nm, and a Gatan 10 M pixel charge-coupled device camera.

#### 2.4.3. XRD

The GNP XRD data, as described in a previous study [32], were used. A Rigaku Benchtop Miniflex XRD (λ_Cu–Kα_ 0.1541 nm, 40 kV, 40 mA, 2θ 20–70°, scanning speed 0.02°/min at 5°/min, room temperature) was used to determine the ECNF purity and structure. FullProf software (version 7.20) was used for Rietveld pattern analysis with a goodness-of-fit from the weighted pattern R-factor (R_wp_), derived Bragg R-factors (R_P_), and expected R-factor (R_exp_). The GNP crystalline phase abundances were obtained from the pattern background, preferred orientation, optimized peak shape, sample displacement, scale factor, 2θ_0_, and lattice parameters. The crystallography open database and graphite crystal structures (COD 9000046) were used for Rietveld refinements.

## 3. Results and Discussion

### 3.1. Microstructure Imaging

Representative low-magnification carbonized morphologies and high-magnification electrospun CNFs are shown by typical secondary electron images in Figure 2a,b, respectively. The ECNFs were uniform with small variations in mean diameter (129 ± 43 nm standard deviation for 50 nanofibers). No microscopically identifiable beads or beaded-nanofibers were identified.

An EDS spectrum for the ECNFs is provided in Figure 3, with a strong C signature (100% C k) from the GNPs/PAN.

Representative low-resolution, granular high-resolution, and lattice–resolved ECNF microstructures as provided by TEM and HR-TEM are given in Figure 4a–c. Polycrystalline ECNFs had a d-spacing between adjacent (002) with ~0.33 nm lattice planes. The nanofiber diameters (104 ± 23 nm) agreed with the SEM results. A skin-core structure in the fiber cross-section and conical platelets were visible. Graphene layers in a seamless cylinder with planes parallel to the longitudinal axis were visible in the ECNFs, and perpendicular layers were present according to the graphene layer stacking. The graphitic ECNFs stacked to form various graphene structures as confirmed by XRD (see Section 3.2).

### 3.2. XRD Results and Rietveld Analysis

Stacked ECNF and GNP XRD spectra are presented in Figure 5. The ECNF XRD and Rietveld analysis follows on from previous GNPs XRD data [32]. The strong diffraction peak at a 2θ of ~26.34° (indexed as (002)) and weaker characteristic diffraction peaks (indexed as (020), (111), and (004) planes) indicate either a hexagonal crystal structure for both samples without impurity or second-phase peaks in the graphitic structure (crystallographic carbon planes).

The crystallite size and peak position from the XRD spectra were used to determine the interlayer spacing. The Scherrer equation was used to obtain the average ECNF crystallite size (DXRD), and Bragg’s Law was used to determine the d_002_ (d-spacing for 2H (002) from a 2θ peak at 26.34°) [32,33,34]:(1)$$D_{XRD}=\frac{k\lambda }{\beta cos\theta }\;$$
(2)$$d_{002}=\frac{n\lambda }{2sin\theta }\;$$
where *k*, *θ*, *β*, and *λ* are the shape factor (0.91), Bragg diffraction angle, full width at half maximum (FWHM), and Cu–K_α_ radiation wavelength (0.15419 nm), respectively.

The ECNF crystallite size was 14.92 nm, which is in agreement with a former GNP crystallite size [32] and Gen et al.’s graphite sample sizes [35]. The broad (002)-like reflection origins as interpreted from the uniform interlayer spacing (*d*_002_) are related to layer misalignment with average interlayer spacings. The naturally occurring graphite crystal (*d*_002_) value is 0.335 nm [36]. These results correlate with the TEM analyzes.

The number of layers along the c-axis (*N_c_*) is defined as [37]:(3)$$N_{c}=\frac{D_{XRD}}{d_{002}}$$

Forty-five ECNFs layers exist along the c-axis as determined from the calculated interlayer spacing (*d*_002_) and apparent crystallite size (*D_XRD_*) in the c-direction.

Rietveld pattern fitting with FullProf software was used to analyze the ECNF XRD spectrum and the estimated goodness-of-fit (GOF) was obtained from the derived Bragg R-factors (R_B_), expected R-factor (R_exp_), and weighted pattern R-factor (R_wp_) [32]. The peak shape parameters, preferred orientation, scale factor, lattice parameters, 2θ_0_, pattern background, and sample displacement were optimized in the Rietveld refinement to calculate the ECNF lattice parameters. Figure 6 shows the residual ECNFs-XRD Rietveld plots. Graphite crystal structures (COD 9000046) were used to obtain the refinements. Calculated and measured values are indicated by solid black lines and red circles, respectively; the residual difference between the two is shown by a blue plot, while the (002), (020), and (004) peak positions are shown by green bars. A 1.59 goodness-of-fit (χ^2^) shows an excellent refinement quality (the ideal goodness-of-fit is 1.0) [32,38].

Table 1 compares the ECNF and GNP cell volume and cell parameters with data reported in the literature. Rietveld hexagonal ECNF and GNP lattice parameters a and c were 0.24567 nm and 0.67753 nm, and 0.24461 nm and 0.67254 nm, respectively. The ECNF and GNP cell volumes were 0.07088 nm^3^ and 0.06996 nm^3^, respectively, with the former being slightly higher than the latter and similar to data reported in the literature. The ECNF lattice parameter increase suggests that PAN addition substitutes carbon atoms as dopants in the carbon lattice rather than occupying interstitial sites.

The Rietveld crystal lattice of the GNPs indicated the existence of parallel 2D graphene layers with a translational …ABAB… sequence of tightly bonded sp^2^ hybridized carbon atoms in hexagonal rings. Graphene layer sliding results from covalently bonded layered carbon atoms that are bound by weak Van Der Waals forces to yield a soft lubricating nature in the GNPs. Adjacent graphene layers in graphite (0.336 nm) are separated by half the hexagonal graphite crystallographic spacing (0.673 nm) and compared with the calculated (002) peak 2θ d-spacing of 26.619°.

## 4. Conclusions

Electrospun carbon nanofibers were prepared for the first time from PAN and GNPs. A PAN solution and GNP precursor were ejected through a needle in a high electric field to form composite CNFs. The ECNF microstructure, as studied by SEM and HR-TEM, showed a smooth nanofiber surface (129 ± 43 nm diameter) and stacked graphitic and graphene sheets that formed rolled-shaped structures in a seamless cylinder and a plane that was parallel to the longitudinal axis. Perpendicular layers formed when carbonization was conducted in argon at 500 °C. XRD and Rietveld refinement showed an ECNF crystal structure with a nanometer-thick worm-like GNPs and an estimated graphitic crystallite size of 14.92 nm. The ECNFs serve as a promising low-cost, lightweight structural alternative to carbon- and metal-based electrically conductive–reinforcement applications.

## Figures and Tables

**Figure 1 materials-16-01749-f001:**
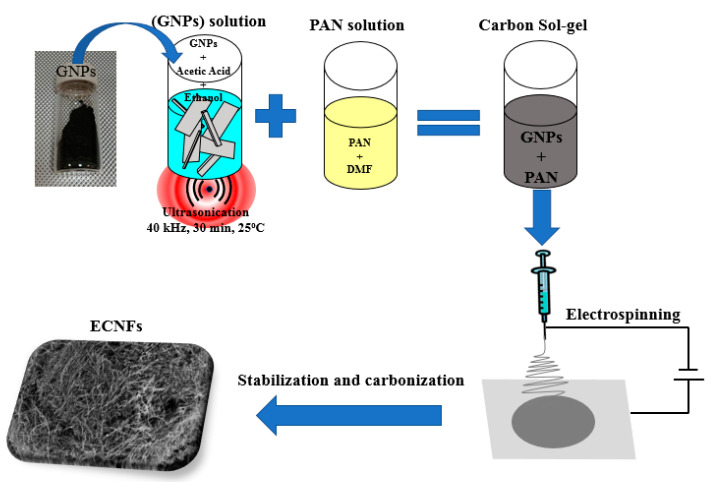
Schematic of CNF synthesis.

**Figure 2 materials-16-01749-f002:**
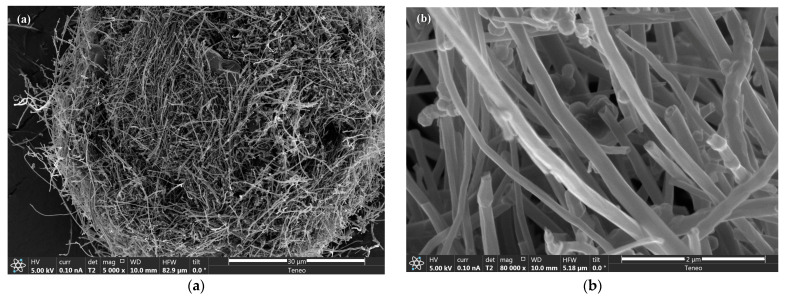
(**a**) Low- and (**b**) high-magnification SEM image of electrospun carbon nanofibers.

**Figure 3 materials-16-01749-f003:**
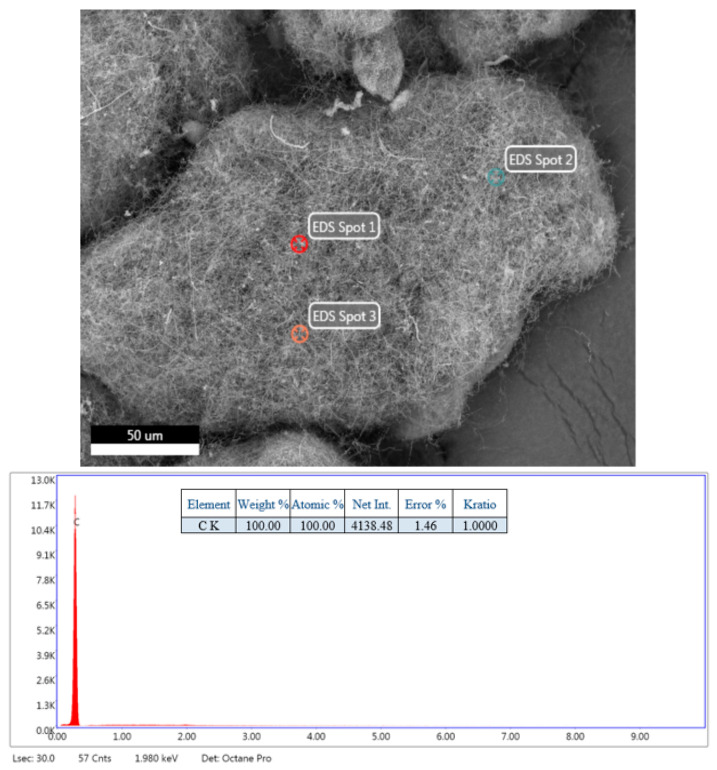
EDS of electrospun carbon nanofibers (ECNFs) after carbonization in argon for 1 h at a peak 500 °C in a single step with a 5 °C/min heating rate.

**Figure 4 materials-16-01749-f004:**
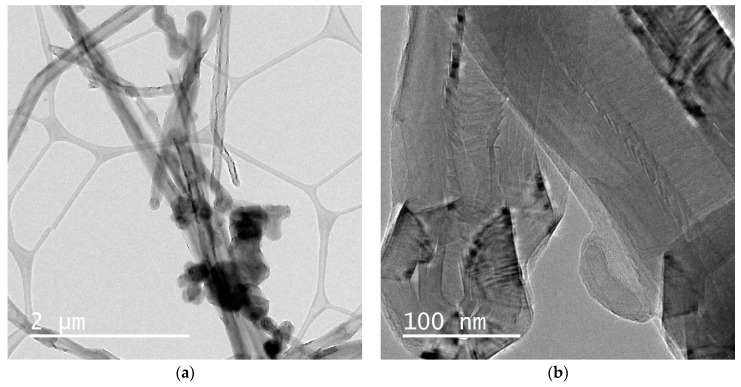
(**a**) Low- and (**b**) high-magnification ECNF TEM images and (**c**) lattice-resolved HRTEM images with polycrystalline carbon material and d-spacing between adjacent (002) and ~0.33 nm lattice planes.

**Figure 5 materials-16-01749-f005:**
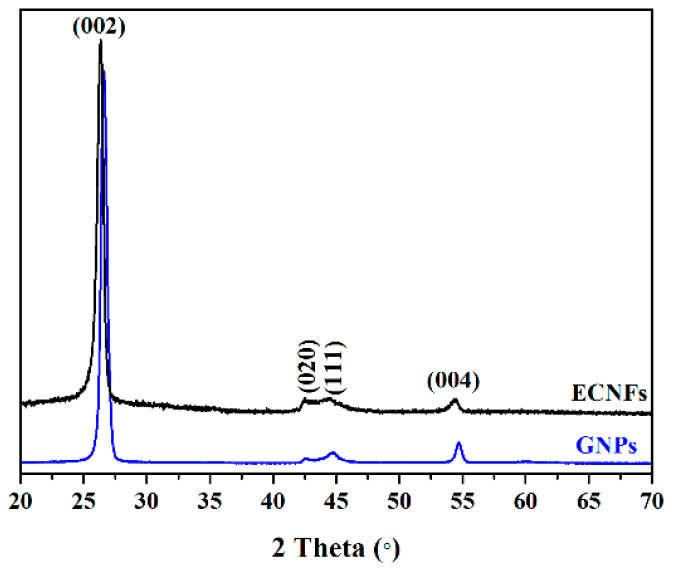
ECNF and GNP stacked XRD spectra.

**Figure 6 materials-16-01749-f006:**
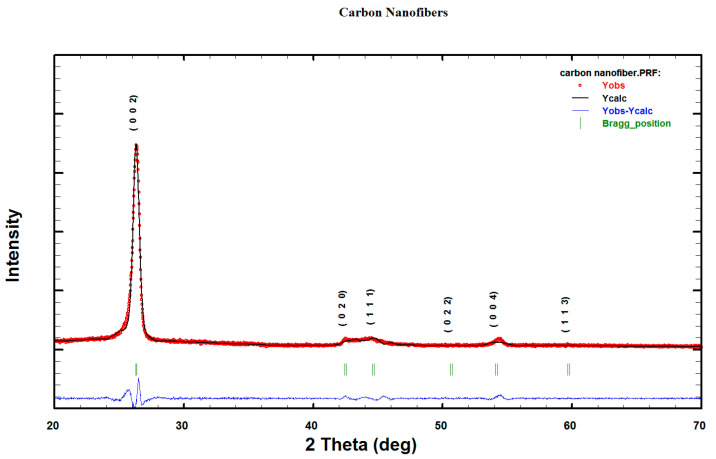
ECNFs XRD Rietveld difference plots. Measured and calculated patterns shown by red circles and solid black line, respectively. Difference between calculated and measured pattern and peak positions ((002), (020), and (004)) shown by blue residual plot and green bars, respectively.

**Table 1 materials-16-01749-t001:** Comparison of hexagonal graphite lattice parameters, cell volume, and density with data reported in the literature.

Study	a (nm)	c (nm)	V (nm^3^) *
ECNFs	0.24567	0.67753	0.07088
[32]	0.24461	0.67254	0.06996
[39]	0.24560	0.66960	0.07081
[40]	0.24600	0.67100	0.07116
[41]	0.24620	0.67110	0.07133
[42]	0.24630	0.67120	0.07142
[43,44]	0.24612	0.67080	0.07126

* Cell volume from XRD Rietveld analysis for ECNFs and GNPs and calculated from a hexagonal cylinder volume in other studies.

## Data Availability

All datasets used for this study are included in the manuscript.

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
