# Peer review of "Synthesis and Characterization of Electrospun Carbon Nanofibers from Polyacrylonitrile and Graphite Nanoplatelets"

_materials, 2023, doi:10.3390/ma16041749_

Round 1

Reviewer 1 Report

The authors presented a synthesis and characterization of electrospun carbon nanofibers from polyacrylonitrile and graphite nanoplatelets by using sol-gel electrospinning process. The presented results are interesting and significant. This manuscript is recommended to be published after minor revision. The suggested changes are as follows:

1.     What are other techniques to synthesis electrospun carbon nanofibers?

2.     Authors should mention the advantages of sol-gel electrospinning techniques over other methods.

3.     In figure 2. (a) Low and (b) high-magnification SEM image of electrospun carbon nanofibers, the magnification scale label should be enlarge.

4.     Authors should include applications of synthesis of electrospun carbon nanofibers which are synthesized using polyacrylonitrile and graphite nanoplatelets.

Author Response

First of all, I highly appreciate you for giving us the opportunity to revise and resubmit the manuscript (materials-2147224) entitled “Synthesis and characterization of electrospun carbon nano-fibres from polyacrylonitrile and graphite nanoplatelets”. I am also very grateful to the reviewers for their constructive suggestions and for their proposed corrections which have undoubtedly improved the manuscript. I have addressed all the issues raised and have revised the manuscript accordingly. The concerns raised by the reviewers have been fully addressed. Below are I itemized responses (in red) to the reviewers’ comments. The revisions are indicated by blue-marked text in the manuscript.

Reviewer 2 Report

In this study, polyacrylonitrile (PAN) and graphite nanoplatelets (GNPs) were used to prepare ECNFs by electrospinning and carbonization. Some comments were given in order to improve the manuscript. 

Experimental methodology

1. What is the mean size of graphite nanoplatelets used? Any SEM image can be provided?

2. Did you examine the functional groups of the electrospun carbon nanofibers by FTIR?

3. “A carbon solution of GNPs was prepared by mixing pulverized and sieved (200 mesh) GNP powder …”. Please check it should be “A carbon suspension” (carbon nanoplatelets suspended in solvent mixture). “solution” => “suspension”.

4. The ECNFs were heated in air from room temperature to 250°C in a furnace for 5 h at 5 °C/min with a 1 h hold time at the peak temperature. What is the purpose of this step? Did you check the ECNFs by EDS if the fibres were oxidized? 

 Results and discussion

1. In Fig. 3, the EDS of ECNFs after carbonization in argon for 1 h at a peak 500°C?

2. In the future study, will you evaluate the mechanical and electrical properties of the ECNFs?  

Author Response

(The authors gave the same response as above.)
